# Farmer Participatory Evaluation of Sorghum Varieties in Flood Recession Agriculture Systems in North-Western Mali



**Kalifa Traore [1], Bouya Traore [2], Abdoulaye Diallo [2], Gry Synnevag [3] and Jens B. Aune [3,***

[1] Institute of Rurale Economie (IER), Scientific Director, Rue Mohamed V Quartier du Fleuve, Bamako BP262, Mali; ibosimon_1@yahoo.fr

[2] Institute of Rural Economy (IER), Regional Center of Agronomic Research of Sotuba/Sorghum Program, Bamako BP262, Mali; bouyatr1@gmail.com (B.T.); dialloabdoulayeier@gmail.com (A.D.)

[3] Department of International Environment and Development Studies (Noragric), Faculty of Landscape and Society, Norwegian University of Life Sciences, P.O. Box 5003, N-1432 Aas, Norway; gry.synnevag@nmbu.no

* Correspondence: jens.aune@nmbu.no

**Abstract:** Flood recession farming is an important cropping system for ensuring food security in western Mali. The present study identified sorghum varieties adapted to this farming system. In the first year, numerous varieties were tested in the fields of 12 farmers. The 22 best-performing varieties, based on farmers' scores using a preference index (PI), were further studied the following year. In the third year, the four varieties with the highest PI scores were tested against the local variety, Samé. The best-performing varieties were given the names Yélimané 1, Yélimané 2, Yélimané 3, and Yélimané 4. Across the three years, the best-performing variety, Yélimané 1, showed a 60.2% and 55.3% greater grain and stover yield, respectively, compared to the local Samé variety. The four improved varieties also reached maturity 30 days sooner than the local variety. A survey involving 101 farmers showed that the improved varieties, combined with higher plant density, seed priming and microdosing of mineral fertilizer, reduced the number of food-insecure months by 3.59 months. These varieties combined with improved agronomic practices have the potential to improve food security in flood recession areas in West Africa.

**Keywords:** participatory sorghum breeding; yield level; time to maturity; post-harvesting properties; adoption; food security





## 1. Introduction

Flood recession agriculture depends on residual soil moisture that is stored in the soil profile after water recedes from annually inundated flood plains, temporary lakes, or seasonal wetlands [1]. Farmers sow as the water recedes from riverbanks and temporarily flooded lakes [2]. The crops grow to maturity based on the water that is stored in the soil profile [3].

In the Yélimané district in north-western Mali, flood recession farming is practiced in the temporary lakes Terekole, Kolombine, and Magui. These lakes are flooded due to precipitation directly on the floodplain and runoff from areas at higher elevation in the watershed [1]. The lake system constitutes the umbilical cord for crop and livestock production in the area. Studies conducted by the Groupe de Recherche et de Réalisation pour le Développement Rural (GRDR) have estimated that 70,000 ha of land could potentially be used for flood recession farming in the Kayes region [4]. The area that can be used for flood recession farming is normally larger in years with ample rainfall. The length of the flooding period also matters; if the flooding period is short, the soil will not be sufficiently saturated with water [1,5]. In the Tombouctou region, a different type of flood recession farming is practiced. Here, sorghum is sown in February and March after the water recedes from the riverbanks of the Niger. The growing cycle is completed from June to September

depending on the rainy season. The sorghum varieties used in the Tombouctou region are therefore late-maturing varieties.

Mali faces great challenges to feed its population, which is growing at a rate of 3% per year. Technologies that can increase food production such as improved crop varieties are therefore in high demand. Not much research has been conducted on flood recession farming in Mali, but the project "Adapting Crop and Livestock Systems to Climate Change" has addressed this situation by developing improved cultivation methods and through varietal development.

In the Yélimané district, there is a strong complementarity between rainfed and flood recession farming. In years with erratic rainfall, rainfed cropping often fails, but flooding may still take place, allowing farmers to secure food production based on flood recession farming. There is a particular need to develop crop varieties that match the growing season in flood recession farming. The water storage capacity of soil is limited, and early maturing varieties are therefore of great interest [1].

In the Yélimané district, annual sorghum production was estimated at 14,025 tons against 1,250,868 tons nationwide [5,6]. Sorghum is a staple food crop for millions of Malian smallholder farmers, and thus plays an important role in achieving food security [7]. However, only 32% of farmers in Mali use improved varieties of this plant [8]. Farmers may be unaware of new varieties, or new varieties may not correspond to their preferences regarding processing properties and taste [1]. Thus, farmers need to test a range of varieties under their own conditions, resource levels, and environment to select the ones they prefer [9]. Consequently, learning from and interacting with farmers is important to improve the suitability of varieties and thus improve the rate of adoption [8].

Sorghum grain can be used for different purposes, according to the texture of the grains. Grain with a hard texture can be used in thick porridges (tô) or couscous; grain of an intermediate texture can be used in unfermented bread or boiled (as rice); and soft grains are used in fermented breads [10]. Such usage of sorghum grains also exists in Yélimané.

Flood recession farming is a production system that has not been widely studied in Mali, and sorghum breeding has not previously been undertaken to develop varieties which are particularly suitable for flood recession farming. The objectives of this research were to: (a) characterize local sorghum varieties used in flood recession farming; (b) compare the yield levels of introduced improved and local varieties in farmers' fields; (c) assess the relationship between the different characteristics of sorghum varieties; (d) assess farmers' preferences regarding sorghum varieties; and (e) assess the effect of a production package including improved varieties on food security in the Yélimané area. The overall objective of the project was to identify sorghum varieties for flood recession farming in order to improve food security.

## 2. Materials and Methods

### 2.1. Study Sites

The study was undertaken in the Yélimané district (15°3′52″ North, 10°33′57″ West) in the Kayes region of north-western Mali (Figure 1).

The area received between 500 to 600 mm of rainfall per year during the study period. Rainfall is unimodal and highly variable during the year, with the maximum rainfall events occurring in July and August. The average minimum monthly temperatures during the year ranged from 20.2 to 28.5 °C, while the maximum monthly temperatures ranged from 32.9 to 42.5 °C. Sowing takes place at the end of the rainy season (October/November), when the water starts to recede. Sorghum is the major staple crop in flood recession farming in Yélimané.

The dominant soil types are hydromorphic flood plain with gley and pseudogley hydromorphic soils [11]. The cropped zones are mostly flat plains, with soils suitable for flood recession cropping. Soil humidity during the season from 5 October to 25 January was measured every 10th day at depths of 10, 20, 40, 40 and 100 cm using a time-domain reflectometry (TDR) moisture sensor.

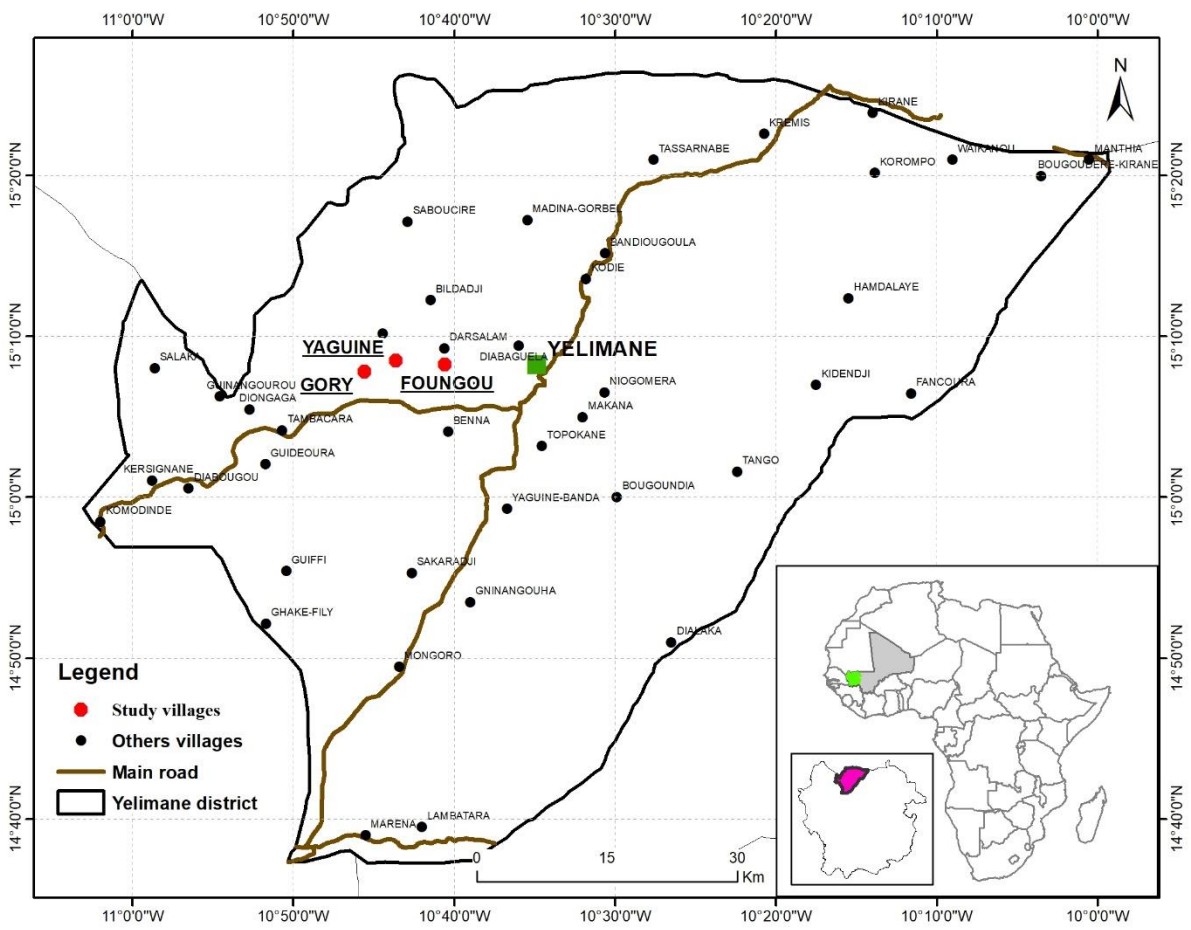

**Figure 1.** Map of Yelimané.

*2.2. Farmer Participatory Evaluation of Sorghum Varieties*

A focus group consisting of eight men and six women, all experienced flood recession farmers, was used to obtain farmers' descriptions of the varieties used in the Yélimané area.

Multi-location replicated farmer participatory variety evaluation trials were conducted in the farmers' fields. The improved varieties were sourced from the sorghum breeding program of IER, while the local varieties were collected from farmers in Yélimané and Tombouctou. These local varieties were crossed with Niètitiama, an improved variety obtained from a combination of the Caudatum and Guinea sorghum types. All the improved varieties have parents collected from varieties used in flood recession farming in Tombouctou or Yélimané. The varieties were given names according to the local language. Saba means sorghum of the Guinea type, while soto and tienda, respectively, mean early (90 days) and late (120 days) maturing varieties.

The study used a three-stage process for participatory testing and evaluation of the sorghum varieties. In the years 2013/2014 and 2014/2015, the varieties were tested in the Gory, Yaguiné, and Foungou villages, and in each of the villages, four farmers served as replicates (12 farmers each year in total). The varieties were assessed by the farmers using a preference index (PI). In the first year, 54 varieties were tested, of which 10 were local varieties. The 22 varieties with the highest PI score in the 2013/2014 season were included in the experiment in the 2014/2015 season. In the final year (2015/2016), the four best varieties in 2014/2015, according to the PI, were compared with the local variety, Samé, in eight villages in the Yélimané district. In each village, eight farmers served as replicates (64 farmers in total). Farm management at the sites was according to farmers' practices. Moreover, the characteristics of the local varieties used were collected from farmers to better understand their preferences regarding agronomic factors (growth cycle, grain size,

yield) and organoleptic characteristics (color, taste, diversity of dishes etc.). Some of the information given by the farmers, such as cycle and grain and straw yield were validated by the results collected in on-farm trials.

The experiments were conducted using a randomized block design where a farmer represented a replication. The varieties were sown at a density of 50,000 plants ha$^{-1}$ (one meter between rows and 40 cm between hills in the row), and each hill was thinned to two plants per hill. Fertilization consisted of 50 kg ha$^{-1}$ of urea, 25 kg ha$^{-1}$ applied at sowing and 25 kg ha$^{-1}$ applied 10 days after sowing. The elementary plot measured 1.6 m $\times$ 5 m (8 m$^2$). Sowing was carried out on two 5-m long rows of per variety.

2.2.1. Farmer Participatory Evaluation of Sorghum Varieties Using a Preference Index

At the end of the first cropping season (at harvesting period), the first participatory evaluation of different varieties was undertaken by asking farmers to rank the varieties through a voting system. The ranking was undertaken separately for men and women farmers.

The ranking was done by placing large non-transparent envelopes (40 cm $\times$ 30 cm) in front of each plot. Cards with three different colors were given to each farmer. Each farmer had three colors, with the number of each color being equal to the number of varieties to be evaluated. Ranking of the varieties was undertaken by 17, 27, and 20 female farmers from the Foungou, Gory, and Yaguine villages, respectively, and 15, 15, and 17 male farmers from the same villages. These farmers were volunteers, approved by villagers, based on their interest in testing varieties in their own fields. When a farmer preferred a variety, a green card was put in the envelope. A yellow card was given if there was any hesitation and the variety needed to be re-evaluated in the following cropping season. Finally, a red card was put in the envelope when a variety was rejected. Then, the number (N) of green, yellow, and red cards was counted per variety and the PI was separately calculated for men and women. The formula described in [12] was used:

$$\text{PI } (\%) = \frac{100 \times \left[\text{N}_{\text{green}} + 0.5 \times \text{ yellow}\right]}{\text{Total number of voting participants in the target group}}$$

The PI ranged from 0% to 100%, with 0% meaning complete rejection and 100% meaning approval by all participants. In addition, interviews with individual farmers (female and male) were undertaken to collect the reasons for their variety choice or rejection.

2.2.2. Assessing the Combined Effect of Improved Varieties and Improved Production Methods on Food Security

The effect of an introduced production method on food security was assessed. The package consisted of improved varieties, microdosing (organic and mineral fertilizer), line planting, improved density (1 m $\times$ 0.5 m instead of 1 m $\times$ 1 m), seed priming (about 80% of the farmers used seed priming), and seed treatment with a combined insecticide/fungicide. The farmers adopted the elements of the package that they found the most interesting. The effect of the introduced package was assessed by asking the farmers to compare the number of food-insecure months before and after the new technology was introduced. The months during which household heads had enough food to satisfy their family's needs were considered food-secure months. In total, 101 households participated in the survey. The farmers selected for the questionnaire were the household heads from the villages of Foungou, Gory and Yaguine. The percentage of improved varieties used by household heads was determined as follows:

$$\text{UIV} = \frac{NU}{TN} \times 100 \tag{1}$$

where UIV is the use of improved varieties (%); *NU* is the number of users of improved varieties; and *TN* the total number of farmers.



### 2.2.3. Statistical Analysis

An ANOVA analysis (complete randomized block design) was undertaken using the MINITAB-18 statistical software. The effects of the treatments were considered significant at the probability threshold of $p < 0.05$. The Newman–Keuls test was used for separation of the treatment means. Descriptive statistics were used to compare respondent frequencies with respect to farmers' variety choices and the reasons thereof.

## 3. Results

Results are presented regarding the characteristics of traditional sorghum varieties, yield performance of local and improved varieties, farmers' preferences of varieties according to gender, the correlation between different characteristics of sorghum varieties, and the effect of a production package including improved varieties on food security.

### 3.1. Characteristics of Farmers' Varieties

A characterization of the varieties used in flood recession farming in Yélimané and Tombouctou showed that the varieties differed greatly regarding time to maturity, grain and straw yield, grain color, grain size, and taste (Table 1). Time to maturity varied from 75 to 180 days. The varieties originating from the Tombouctou region had a growing cycle of up to 180 days, and these varieties had a particularly good straw yield. Most of the varieties had a white grain color. Grains were used for preparing couscous, tô, or porridge, or were boiled as rice. As can be seen in Table 1, not all varieties had a good taste; the local varieties Magalèmè and Samé had an excellent taste. It is important to be aware that some of the local varieties in Table 1 may be identical, even though they have different names, as farmers may give local names that are specific to their village. In the Yélimané area, "Samé" and "Magalèmè" were the most common varieties.

**Table 1.** Farmers' characterization of sorghum varieties in flood recession area of Yélimané.

| Variety Characteristics | Growth Cycle | Grain Yield (kg ha$^{-1}$) | Straw Yield (kg ha$^{-1}$) | Grain Color | Grain Size | Preferred Food Preparation and Taste |
|---|---|---|---|---|---|---|
| Modigiby (local Yélimané) | 170–180 days | 300 | 2000 | White | small | Couscous |
| Samba-nouha (local Yélimané) Diafouno, Gory, Toya; Kersigueni, Djankoulaga, Tambacara, Toya | 90 days | 400–500 | 2000 | Pale white | Small and vitreous grain | Couscous and tô (solid starch made from pounded cereal) |
| Samé (local Yélimané) | 90 days | 400–500 | 2000 | Pale white | Small and vitreous grain | Good tasting couscous and tô, Highly palatable for livestock |
| Boydibo (local Yélimané) | 90 days | 400 | 1500–1800 Palatable to livestock | White | Very big but mature in the hull | Bad taste for couscous and tô; Porridge and cooked as rice rated as very bad |
| Motafara (local Tombouctou) | 130–150 months | 350–400 | Palatable to livestock | White | Small | Only porridge and cooked as rice had very good taste |
| Saba tienda (local Tombouctou) | 150 days | 300 | High tillering ability | Scarlet White | intermediate | Good taste but only for couscous and tô |
| Lagahiré (local Yélimané) = white Lagahiré | 75 days | 400–500 | 1500 Palatable to livestock and used as sugar cane | White | Big grain and highly appreciated by birds | Good taste couscous and tô |
| Magalèmè (local Yélimané) | 90 days | 600 | 1800–2000 Highly Palatable to livestock | White | Intermediate and vitreous grain | All the meals are delicious; even the bran is delicious |
| Lagahiré doumbé (local Yélimané) | 75 days | | 1500 | Red but become white when dehulled | Big grain size and highly appreciated by birds | Only couscous and tô are satisfactory Seri and rice are very bad |

### 3.2. Performance of Traditional and Improved Sorghum Varieties

The varieties were tested in the 2013/2014, 2014/2015, and 2015/2016 seasons. The grain yield for the 54 varieties tested in 2013/2014 varied from 422 to 1178 kg ha$^{-1}$, with an average yield of 797 kg ha$^{-1}$ (Table 2). The top five performing varieties were Saba soto-23, Saba soto-32-1, Saba tienda-8, Saba tienda-29, and Modigiby (local Yélimané). These varieties all gave a yield above 1,050 kg ha$^{-1}$, while the local variety, Samé, had a yield of 994 kg ha$^{-1}$.

**Table 2.** Average sorghum grain yield (kg ha$^{-1}$) across sites of flood recession farming for 54 varieties tested in Yaguine, Gory, and Foungou in the 2013/2014 cropping season.

| Varieties | Mean | Separation of Means |||||||||||||||
|---|---|---|---|---|---|---|---|---|---|---|---|---|---|---|---|---|
| Saba soto-23 | 1178 | A | | | | | | | | | | | | | | |
| Saba soto-32-1 | 1125 | A | B | | | | | | | | | | | | | |
| Saba tienda-8 | 1091 | A | B | C | | | | | | | | | | | | |
| Saba tienda-29 | 1059 | A | B | C | D | | | | | | | | | | | |
| Modigiby (local Yélimané) | 1050 | A | B | C | D | | | | | | | | | | | |
| Saba tienda-39 | 1013 | A | B | C | D | E | | | | | | | | | | |
| Samba-nouha (local Yélimané) | 1006 | A | B | C | D | E | | | | | | | | | | |
| Saba soto-27 | 997 | A | B | C | D | E | F | | | | | | | | | |
| Saba soto-17 | 997 | A | B | C | D | E | F | | | | | | | | | |
| **Samé (local Yélimané)** | 994 | A | B | C | D | E | F | G | | | | | | | | |
| Saba tienda-37 | 966 | A | B | C | D | E | F | G | H | | | | | | | |
| Saba soto-35 | 956 | A | B | C | D | E | F | G | H | I | | | | | | |
| Saba tienda-36 | 925 | A | B | C | D | E | F | G | H | I | J | | | | | |
| 82B | 922 | A | B | C | D | E | F | G | H | I | J | K | | | | |
| Boydibo (local Yélimané) | 894 | | B | C | D | E | F | G | H | I | J | K | | | | |
| Saba tienda-33 | 875 | | B | C | D | E | F | G | H | I | J | K | L | | | |
| Saba soto-24 | 875 | | B | C | D | E | F | G | H | I | J | K | L | | | |
| Motafara (local Tombouctou) | 869 | | B | C | D | E | F | G | H | I | J | K | L | | | |
| Saba soto-20 | 859 | | B | C | D | E | F | G | H | I | J | K | L | | | |
| Saba soto-32-2 | 844 | | B | C | D | E | F | G | H | I | J | K | L | M | | |
| 3009B | 831 | | | C | D | E | F | G | H | I | J | K | L | M | | |
| Saba soto-10 | 825 | | | C | D | E | F | G | H | I | J | K | L | M | | |
| Saba tienda-32 | 822 | | | C | D | E | F | G | H | I | J | K | L | M | | |
| Saba soto-5 | 822 | | | C | D | E | F | G | H | I | J | K | L | M | | |
| Saba tienda (local Tombouctou) | 809 | | | C | D | E | F | G | H | I | J | K | L | M | N | |
| Saba soto-22 | 809 | | | C | D | E | F | G | H | I | J | K | L | M | N | |
| Saba soto-4 | 797 | | | | D | E | F | G | H | I | J | K | L | M | N | O |
| Saba soto-28 | 791 | | | | D | E | F | G | H | I | J | K | L | M | N | O |
| Saba tienda-34 | 778 | | | | D | E | F | G | H | I | J | K | L | M | N | O |
| Saba soto-3 | 756 | | | | | E | F | G | H | I | J | K | L | M | N | O |
| Lagahiré (local Yélimané) | 753 | | | | | E | F | G | H | I | J | K | L | M | N | O |
| 12B | 750 | | | | | E | F | G | H | I | J | K | L | M | N | O |
| Saba tienda-30 | 747 | | | | | E | F | G | H | I | J | K | L | M | N | O |
| Saba soto-26 | 722 | | | | | | F | G | H | I | J | K | L | M | N | O |
| Saba soto-19 | 713 | | | | | | | G | H | I | J | K | L | M | N | O |
| Saba soto-13 | 706 | | | | | | | | H | I | J | K | L | M | N | O |
| Saba soto-16 | 700 | | | | | | | | H | I | J | K | L | M | N | O | P |
| Saba soto-1 | 697 | | | | | | | | H | I | J | K | L | M | N | O | P |
| Saba tienda-38 | 684 | | | | | | | | H | I | J | K | L | M | N | O | P |
| Malisor 92-1 | 681 | | | | | | | | | I | J | K | L | M | N | O | P |
| Saba soto-6 | 675 | | | | | | | | | I | J | K | L | M | N | O | P |
| Saba soto (local Tombouctou) | 672 | | | | | | | | | | J | K | L | M | N | O | P |
| Saba soto-21 | 650 | | | | | | | | | | J | K | L | M | N | O | P |
| Saba soto-18 | 644 | | | | | | | | | | J | K | L | M | N | O | P |

**Table 2.** *Cont.*

| Varieties | Mean | Separation of Means | | | | | | |
|---|---|---|---|---|---|---|---|---|
| Magalèmè (local Yélimané) | 641 | K | L | M | N | O | P | |
| Saba soto-12 | 606 | | L | M | N | O | P | |
| Saba soto-25 | 600 | | L | M | N | O | P | |
| Saba soto-14 | 594 | | L | M | N | O | P | |
| Saba soto-9 | 569 | | | M | N | O | P | |
| CSM 63E | 538 | | | | N | O | P | |
| Saba soto-11 | 538 | | | | N | O | P | |
| Saba tienda-31 | 516 | | | | | O | P | |
| Lagahiré doumbé (local Yélimané) | 422 | | | | | | P | |
| Saba soto-15 | 422 | | | | | | P | |
| Mean | 792 | | | | | | | |
| SD | 203 | | | | | | | |
| CV (%) | 25.7 | | | | | | | |

A, B, C, D, E, F, G, H, I, J, K, L, M, N, O, P: values in the Samé column with different letters are significantly different at $p$ = 0.05. SD: standard deviation; CV: coefficient of variation (this cannot be correct). It should be (varieties that do not have overlapping letters are significantly different).

As Figure 2 shows, the best yield performing varieties had the shortest growing season ($R^2$ = 0.69). The time from sowing to maturity varied from 90 to 128 days for the varieties tested in the 2013/2014 season.

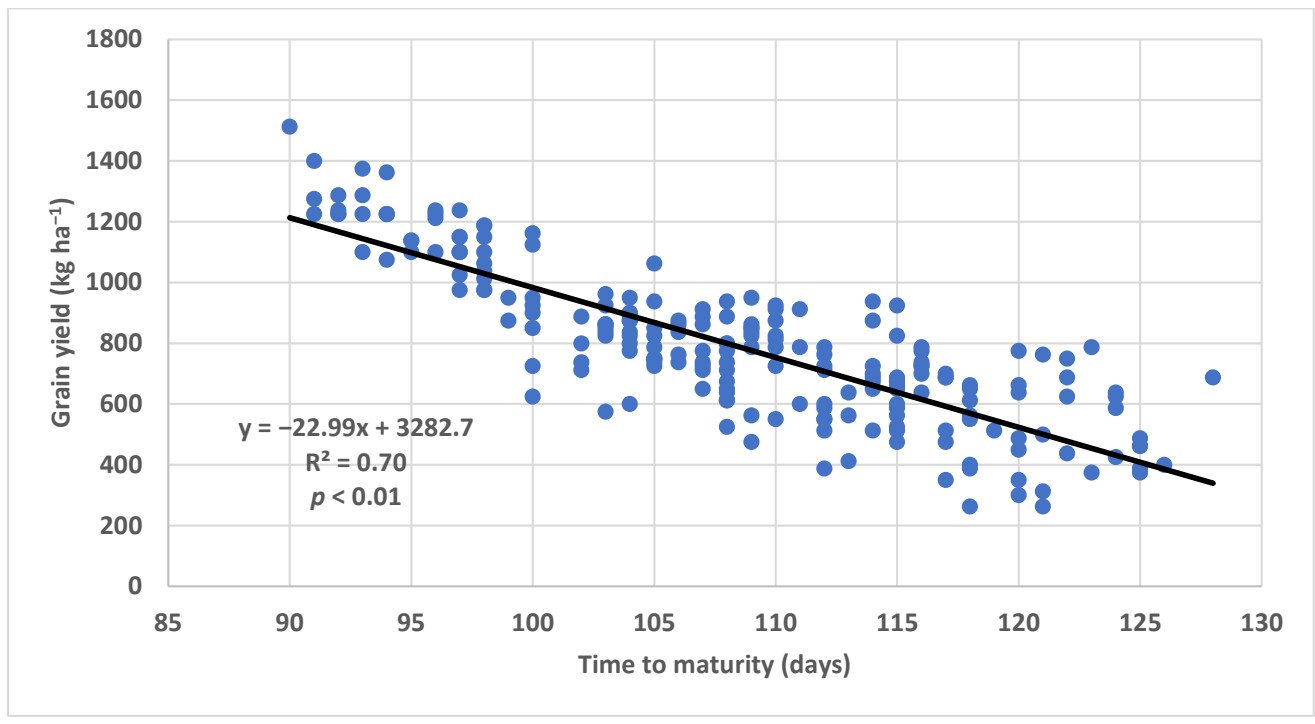

**Figure 2.** Linear regression of the yields of 54 sorghum varieties and time to maturity (2013/2014 season, Yélimané).

Straw yield for the varieties tested in 2013/2014 ranged from 1844 kg ha$^{-1}$ to 9906 kg ha$^{-1}$, with an average of 3379 kg ha$^{-1}$ (Table 3). The variety Saba soto-23 had the highest yield (9906 kg ha$^{-1}$).

**Table 3.** Average sorghum straw yield (kg ha$^{-1}$) across sites of flood recession farming for 54 varieties tested in Yaguine, Gory, and Foungou in the 2013/2014 cropping season. Varieties with letters that do not overlap (A to M) are significantly different.

| Varieties | Mean | A | B | C | D | E | F | G | H | I | J | K | L | M |
|---|---|---|---|---|---|---|---|---|---|---|---|---|---|---|
| Saba soto-23 | 9906 | A | | | | | | | | | | | | |
| Saba soto-8 | 7750 | | B | | | | | | | | | | | |
| Malisor 92-1 | 6781 | | B | C | | | | | | | | | | |
| Lagahiré (local Yélimané) | 6375 | | | C | | | | | | | | | | |
| Saba tienda-32 | 6281 | | | C | | | | | | | | | | |
| Samé (local Yélimané) | 5688 | | | C | | | | | | | | | | |
| Saba tienda-35 | 4219 | | | | D | | | | | | | | | |
| Saba soto-15 | 3750 | | | | D | E | | | | | | | | |
| Saba tienda-29 | 3719 | | | | D | E | | | | | | | | |
| Saba soto-17 | 3656 | | | | D | E | F | | | | | | | |
| Saba soto-27 | 3625 | | | | D | E | F | G | | | | | | |
| 3009B | 3593 | | | | D | E | F | G | H | | | | | |
| Saba soto-25 | 3563 | | | | D | E | F | G | H | | | | | |
| Saba tienda-37 | 3500 | | | | D | E | F | G | H | I | | | | |
| Saba soto-7 | 3375 | | | | D | E | F | G | H | I | J | | | |
| Saba soto-24 | 3375 | | | | D | E | F | G | H | I | J | | | |
| Modigiby (local Yélimané) | 3344 | | | | D | E | F | G | H | I | J | | | |
| Saba tienda-36 | 3313 | | | | D | E | F | G | H | I | J | K | | |
| Saba tienda (local Tombouctou) | 3281 | | | | D | E | F | G | H | I | J | K | L | |
| Saba tienda-39 | 3281 | | | | D | E | F | G | H | I | J | K | L | |
| Samba-nouha (local Yélimané) | 3250 | | | | D | E | F | G | H | I | J | K | L | |
| 82B | 3159 | | | | D | E | F | G | H | I | J | K | L | |
| Saba soto-5 | 3094 | | | | D | E | F | G | H | I | J | K | L | |
| Saba soto (local Tombouctou) | 3063 | | | | D | E | F | G | H | I | J | K | L | |
| Saba tienda-33 | 3031 | | | | | E | F | G | H | I | J | K | L | |
| Saba soto-2 | 3031 | | | | | E | F | G | H | I | J | K | L | |
| Saba soto-26 | 3000 | | | | | E | F | G | H | I | J | K | L | M |
| Saba tienda-34 | 2969 | | | | | E | F | G | H | I | J | K | L | M |
| Saba soto-6 | 2969 | | | | | E | F | G | H | I | J | K | L | M |
| Saba soto-16 | 2906 | | | | | E | F | G | H | I | J | K | L | M |
| Motafara (local Tombouctou) | 2813 | | | | | E | F | G | H | I | J | K | L | M |
| Boydibo (local Yélimané) | 2813 | | | | | E | F | G | H | I | J | K | L | M |
| 12B | 2781 | | | | | E | F | G | H | I | J | K | L | M |
| Saba soto-3 | 2781 | | | | | E | F | G | H | I | J | K | L | M |
| Saba soto-22 | 2781 | | | | | E | F | G | H | I | J | K | L | M |
| Saba soto-4 | 2750 | | | | | E | F | G | H | I | J | K | L | M |
| Saba soto-20 | 2750 | | | | | E | F | G | H | I | J | K | L | M |
| Saba tienda-30 | 2656 | | | | | E | F | G | H | I | J | K | L | M |
| Saba soto-1 | 2594 | | | | | E | F | G | H | I | J | K | L | M |
| Magalèmè (local Yélimané) | 2531 | | | | | | F | G | H | I | J | K | L | M |
| Saba tienda-31 | 2531 | | | | | | F | G | H | I | J | K | L | M |
| Saba soto-28 | 2531 | | | | | | F | G | H | I | J | K | L | M |
| Saba soto-10 | 2531 | | | | | | F | G | H | I | J | K | L | M |
| Saba soto-21 | 2500 | | | | | | F | G | H | I | J | K | L | M |
| Saba soto-18 | 2469 | | | | | | | G | H | I | J | K | L | M |
| CSM 63E | 2444 | | | | | | | | H | I | J | K | L | M |
| Saba tienda-38 | 2375 | | | | | | | | | I | J | K | L | M |
| Saba soto-19 | 2250 | | | | | | | | | | J | K | L | M |
| Saba soto-14 | 2219 | | | | | | | | | | J | K | L | M |
| Saba soto-13 | 2219 | | | | | | | | | | J | K | L | M |

**Table 3.** *Cont.*

| Varieties | Mean | Separation of Means | | | | | | | | | | | |
|---|---|---|---|---|---|---|---|---|---|---|---|---|---|
| Saba soto-9 | 2156 | | | | | | | | | K | L | M |
| Saba soto-12 | 2156 | | | | | | | | | K | L | M |
| Lagahiré doumbé (local Yélimané) | 2125 | | | | | | | | | | L | M |
| Saba soto-11 | 1844 | | | | | | | | | | | M |
| Mean | 3379 | | | | | | | | | | | |
| SD | 841 | | | | | | | | | | | |
| CV (%) | 24.9 | | | | | | | | | | | |

A, B, C, D, E, F, G, H, I, J, K, L, M: values in the Samé column with different letters are significantly different at $p = 0.05$. SD: standard deviation; CV: coefficient of variation.

The 22 varieties with the highest PI score from the 2013/2014 season were included in the experiments in the 2014/2015 season. In this season, the grain yield ranged from 235 to 1609 kg ha$^{-1}$, and the average yield was 686 kg ha$^{-1}$ (Table 4).

**Table 4.** Average sorghum grain yield (kg ha$^{-1}$) across sites of flood recession farming for 22 varieties tested in Yaguine, Gory, and Foungou in the 2014/2015 cropping season. Varieties with letters that do not overlap (A to I) are significantly different, according to a Newman–Keuls test.

| Varieties | Mean | Separation of Means | | | | | | | | |
|---|---|---|---|---|---|---|---|---|---|---|
| Saba soto-23 | 1609 | A | | | | | | | | |
| Saba tienda-32-1 | 1494 | A | B | | | | | | | |
| Saba soto-8 | 1459 | A | B | | | | | | | |
| Saba tienda-29 | 1206 | | B | C | | | | | | |
| Samba-Nouha (local Yélimané) | 895 | | | C | D | | | | | |
| Samé (local Yélimané) | 769 | | | | D | E | | | | |
| Saba soto-25 | 766 | | | | D | E | | | | |
| Lagahiré (local Yélimané) | 750 | | | | D | E | F | | | |
| Magalèmè (local Yélimané) | 743 | | | | D | E | F | | | |
| Motafara (local Tombouctou) | 719 | | | | D | E | F | G | | |
| Saba tienda-37 | 664 | | | | D | E | F | G | H | |
| Boidibo (local Yélimané) | 547 | | | | | E | F | G | H | I |
| Malisor 92-1 | 489 | | | | | E | F | G | H | I |
| Saba soto-9 | 411 | | | | | | F | G | H | I |
| Saba soto-10 | 401 | | | | | | | G | H | I |
| Saba soto-3 | 373 | | | | | | | | H | I |
| Saba tienda-34 | 366 | | | | | | | | H | I |
| Saba tienda-31 | 363 | | | | | | | | H | I |
| Saba soto-5 | 285 | | | | | | | | | I |
| CSM 63[E] | 279 | | | | | | | | | I |
| Saba tienda-32-2 | 265 | | | | | | | | | I |
| 82B | 235 | | | | | | | | | I |
| Mean (kg ha$^{-1}$) | 686 | | | | | | | | | |
| SD (kg ha$^{-1}$) | 241 | | | | | | | | | |
| CV (%) | 35 | | | | | | | | | |

A, B, C, D, E, F, G, H, I: values in the Samé column with different letters are significantly different at $p = 0.05$. SD: standard deviation; CV: coefficient of variation.

The five varieties with the highest grain yield were Saba soto-23, Saba tienda-32-1, Saba soto-32-1, Saba soto-8, Saba tienda-29, and Samba-Nouha (local Yélimané). As in the previous season, the best-performing varieties regarding grain yield had the shortest growing season ($R^2 = 0.75$) (Figure 3).

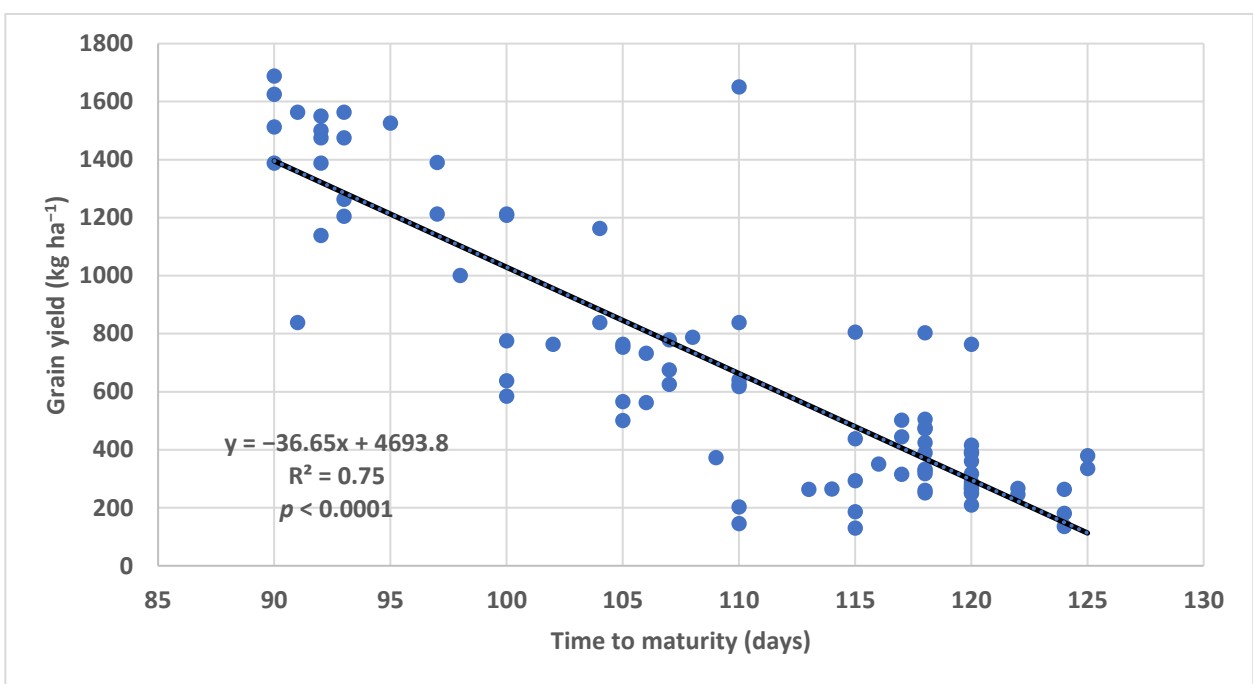

**Figure 3.** Linear regression of the yields of 22 sorghum varieties and time to maturity (2014/2015 season Yélimané).

Straw yield in the 2014/2015 season ranged from 1719 to 4969 kg ha$^{-1}$, with an average of 3284 kg ha$^{-1}$ (Table 5). The varieties with highest grain yield were also those with the highest straw yield.

**Table 5.** Average sorghum straw yield (kg ha$^{-1}$) across sites of flood recession farming for 22 varieties tested in Yaguine, Gory, and Foungou in the 2014/2015 cropping season. Varieties with letters that do not overlap (A to F) are significantly different, according to a Newman–Keuls test.

| Varieties | Mean | Separation Means | | | | | |
|---|---|---|---|---|---|---|---|
| Saba tienda-32-1 | 4969 | A | | | | | |
| Saba soto-8 | 4250 | A | B | | | | |
| Saba soto-23 | 3906 | A | B | C | | | |
| Saba soto-25 | 3781 | A | B | C | D | | |
| Saba tienda-31 | 3625 | | B | C | D | | |
| Saba soto-9 | 3625 | | B | C | D | | |
| 82B | 3531 | | B | C | D | | |
| Samé (local Yélimané) | 3469 | | B | C | D | | |
| Boidibo (local Yélimané) | 3438 | | B | C | D | | |
| Saba soto-10 | 3438 | | B | C | D | | |
| Saba soto-5 | 3344 | | B | C | D | E | |
| Saba tienda-34 | 3313 | | B | C | D | E | |
| Lagahiré (local Yélimané) | 3156 | | B | C | D | E | |
| Malisor 92-1 | 3125 | | B | C | D | E | |
| Saba tienda-32-2 | 3125 | | B | C | D | E | |
| Saba tienda-29 | 3063 | | B | C | D | E | |
| Motafara (local Tombouctou) | 2969 | | | C | D | E | |
| Magalèmè | 2938 | | | C | D | E | F |
| CSM 63E | 2688 | | | C | D | E | F |
| Samba-Nouha | 2625 | | | | D | E | F |
| Saba tienda-37 | 2156 | | | | | E | F |
| Saba soto-3 | 1719 | | | | | | F |
| Mean (kg ha$^{-1}$) | 3284 | | | | | | |
| SD (kg ha$^{-1}$) | 866 | | | | | | |
| CV (%) | 26.3 | | | | | | |

A, B, C, D, E, F: values in the Samé column with different letters are significantly different at *p* < 0.05. SD: standard deviation; CV: coefficient of variation.

In the third season (2015/2016), we compared the top four ranking varieties from the 2014/2015 season with the local variety, Samé. The grain advantage over the local variety was 64.2, 50.6, 49.0, and 48.7% for Saba soto-23, Saba soto-8, Saba tienda-29, and Saba tienda-32, respectively (Table 6).

**Table 6.** Average sorghum grain yield (kg ha$^{-1}$) across sites of flood recession farming for the five tested varieties in Yaguine, Gory, and Foungou in the 2015/2016 cropping season.

| Varieties | Mean | Separation of Means | |
|---|---|---|---|
| Saba soto-23 | 1608 | A | |
| Saba soto-8 | 1475 | A | |
| Saba tienda-29 | 1458 | A | |
| Saba tienda-32-1 | 1456 | A | |
| Samé (local Yélimané) | 979 | | B |
| Mean (kg ha$^{-1}$) | 1395 | | |
| SD (kg ha$^{-1}$) | 165 | | |
| CV (%) | 11.8 | | |

A, B: values in the Samé column with different letters are significantly different at *p* = 0.05. SD: standard deviation; CV: coefficient of variation.

Sabo Soto-23 was the overall best-performing variety across the years regarding grain and straw yield (Table 7). It increased grain yield compared to the local variety (Samé) in the years 2013/2014, 2014/2015, and 2015/2016 by 18.5%, 109.2%, and 64.2% respectively, and the average yield increase across the three years was 60.2%. The percent increase in straw yield for this variety against the Samé variety for these three years were 74.2%, 12.8%, and 68.6%, and the average straw yield increased across the three years by 55.3%.

**Table 7.** Average sorghum straw yield (kg ha$^{-1}$) across sites of flood recession farming for five varieties tested in Yaguine, Gory, and Foungou in the 2015/2016 cropping season.

| Varieties | Mean | Separation of Means | | | |
|---|---|---|---|---|---|
| Saba soto-23 | 5163 | A | | | |
| Saba tienda-32-1 | 4738 | A | B | | |
| Saba soto-8 | 4138 | | B | C | |
| Saba tienda-29 | 3475 | | | C | D |
| Samé (local Yélimané) | 3063 | | | | D |
| Mean (kg ha$^{-1}$) | 4115 | | | | |
| SD (kg ha$^{-1}$) | 667 | | | | |
| CV (%) | 16.2 | | | | |

A, B, C, D: values in the Samé column with different letters are significantly different at *p* = 0.05. SD: standard deviation; CV: coefficient of variation.

These four best preforming varieties were given the names Yélimané 1 (Saba soto-23), Yélimané 2 (Saba soto-8), Yélimané 3 (Saba tiendra 29), and Yélimané 4 (Saba tiendra-32-1). Yélimané 4 reached 50% maturity 65 days after sowing, while the other three improved varieties had a duration of 55 days (CREVV, 2016). The two most commonly grown local varieties (Samé and Magalèmè) needed 90 days to reach 50% maturity.

Figure 4 shows the drying out of the soil profile at 10-day intervals at a depth from 10 cm to 100 cm. The figure shows that after 80 days (between 25 October and 15 January), the amount of water remaining in the soil profile was very low. As such, it is likely that the varieties taking 90 days to reach 50% maturity would have experienced drought at the end of the growing season.

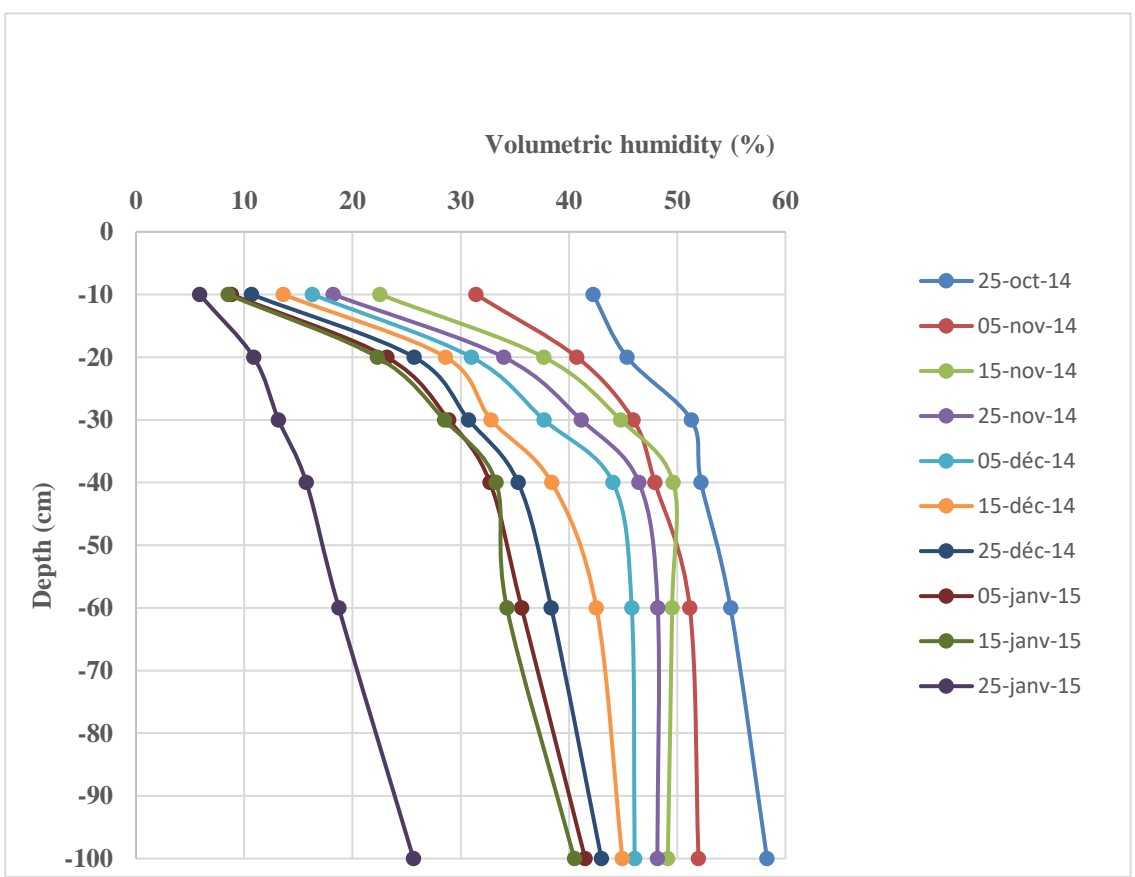

**Figure 4.** Soil moisture dynamic during the 2014/2015 cropping season in Gory and Yélimané.

The improved varieties have white grains, except for Yélimané 1, which has a yellow-colored grain. The 1000-g grain weight for the four varieties are 33.4, 29.6, 26.0, and 21.2 for Yélimané 1, 2, 3, and 4 respectively. The plant heights of the improved varieties are 250, <200, 300, and 300 cm for Yélimané 1, Yélimané 2, Yélimané 3, and Yélimané 4, respectively. The panicles of these four varieties are semi-compact, resembling the Guinea-Caudatum types. A Guinea type was used as the female because of the vitreosity and taste, while a Caudatum type was used as a male to ensure suitable yield characteristics. The Samé variety of the Caudatum type.

*3.3. Relationships between Sorghum Traits among 54 Varieties*

A Pearson correlation analysis was undertaken to assess the relationships between the different characteristics of the sorghum varieties based on data from the 2013/2014 season (Table 8). The characteristics that had the highest correlation coefficient with grain and stover yields were time to 50% flowering and time to maturity. The correlation coefficients between grain yield and time to 50% flowering, and between grain yield and time to maturity, were −0.79 ($p < 0.001$) and −0.84 ($p < 0.001$), respectively, while the correlation coefficient between stover yield and time to 50% flowering and time to maturity was weaker than for grain yield. There was also a significant positive relationship between grain yield and resistance to disease, dehulling properties, grain pounding properties, and taste (Table 8). The correlation coefficient between time to 50% flowering and time to maturity was 0.96 ($p < 0.0001$). There was also a positive correlation coefficient between the post-harvesting properties of dehulling, pounding, and taste.

**Table 8.** Correlation coefficients between yield, days to maturity, resistance to diseases, and processing characteristics of 54 sorghum varieties in Yaguine, Gory, and Foungou in the 2013/2014 cropping season.

|  | R/P | GY | 50%F | DM | SY | PH | RD | DP | GP | T |
|---|---|---|---|---|---|---|---|---|---|---|
| 50%F | R | −0.785 | | | | | | | | |
| | P | 0.0000 | | | | | | | | |
| DM | R | −0.8360 | 0.9590 | | | | | | | |
| | P | 0.0000 | 0.0000 | | | | | | | |
| SY | R | 0.3680 | −0.2760 | −0.3010 | | | | | | |
| | P | 0.0000 | 0.0000 | 0.0000 | | | | | | |
| PH | R | 0.1180 | −0.1730 | −0.1730 | 0.0310 | | | | | |
| | P | 0.0840 | 0.0110 | 0.0110 | 0.6450 | | | | | |
| RD | R | 0.8540 | −0.7130 | −0.7560 | 0.2990 | 0.0440 | | | | |
| | P | 0.0000 | 0.0000 | 0.0000 | 0.0000 | 0.5210 | | | | |
| DP | R | 0.3900 | −0.3750 | −0.3750 | 0.1550 | 0.1670 | 0.3570 | | | |
| | P | 0.0000 | 0.0000 | 0.0000 | 0.0220 | 0.0140 | 0.0000 | | | |
| GP | R | 0.2900 | −0.1590 | −0.1830 | 0.0530 | 0.0360 | 0.2950 | 0.1600 | | |
| | P | 0.0000 | 0.0200 | 0.0070 | 0.4430 | 0.5990 | 0.0000 | 0.0180 | | |
| T | R | 0.3590 | −0.2720 | −0.2920 | 0.1370 | 0.0090 | 0.3240 | 0.0730 | 0.0600 | |
| | P | 0.0000 | 0.0000 | 0.0000 | 0.0440 | 0.8960 | 0.0000 | 0.2850 | 0.3810 | |

R = correlation; P = p value; GY = yield; 50%F = 50% flowering; DM = days to maturity; SY = straw yield; PH = plant height; RD = resistance to diseases; DP = dehulling properties; GP = grain pounding; T = taste.

### 3.4. Farmers' Preference of Sorghum Varieties

The criteria farmers used to select varieties were earliness, grain and straw yield, processing abilities, and taste (Figure 5). Each of these criteria was important for over 80% of the farmers. Grain yield was almost unanimously the most important characteristic for both men and women, with 96% of them rating this property as important. The women tended to give more emphasis to processing properties, seed color and taste than men. Men gave more emphasis to earliness than women. No improved variety was known in the area before the implementation of the variety trials carried out by the project starting in 2013.

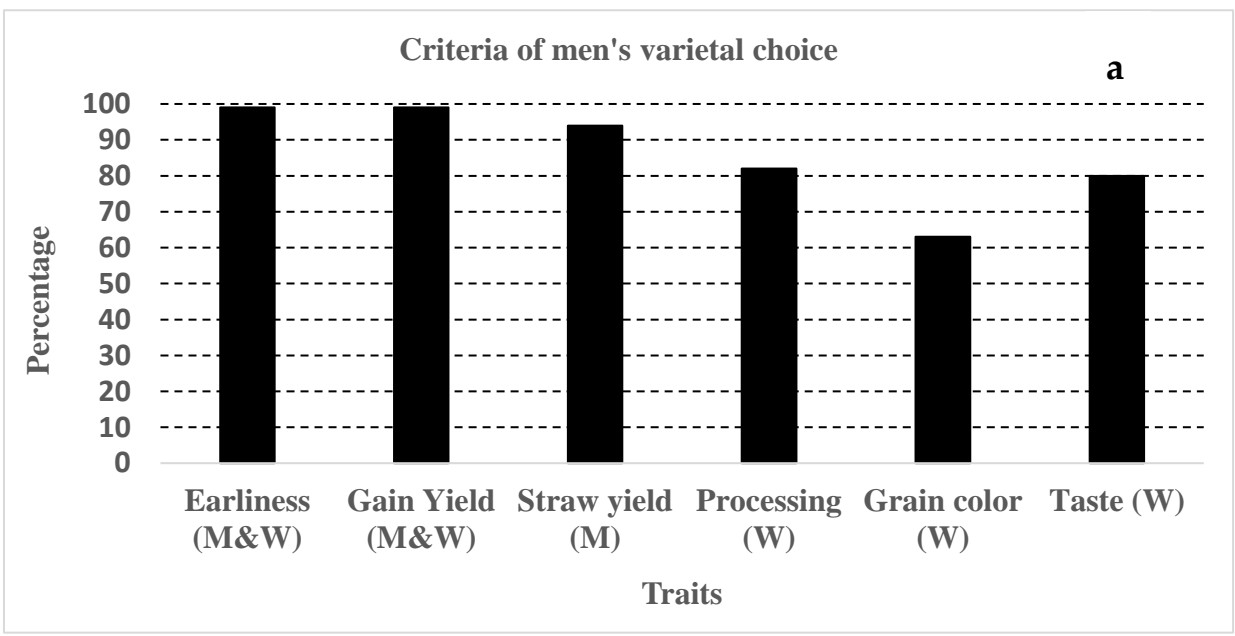

**Figure 5.** *Cont.*

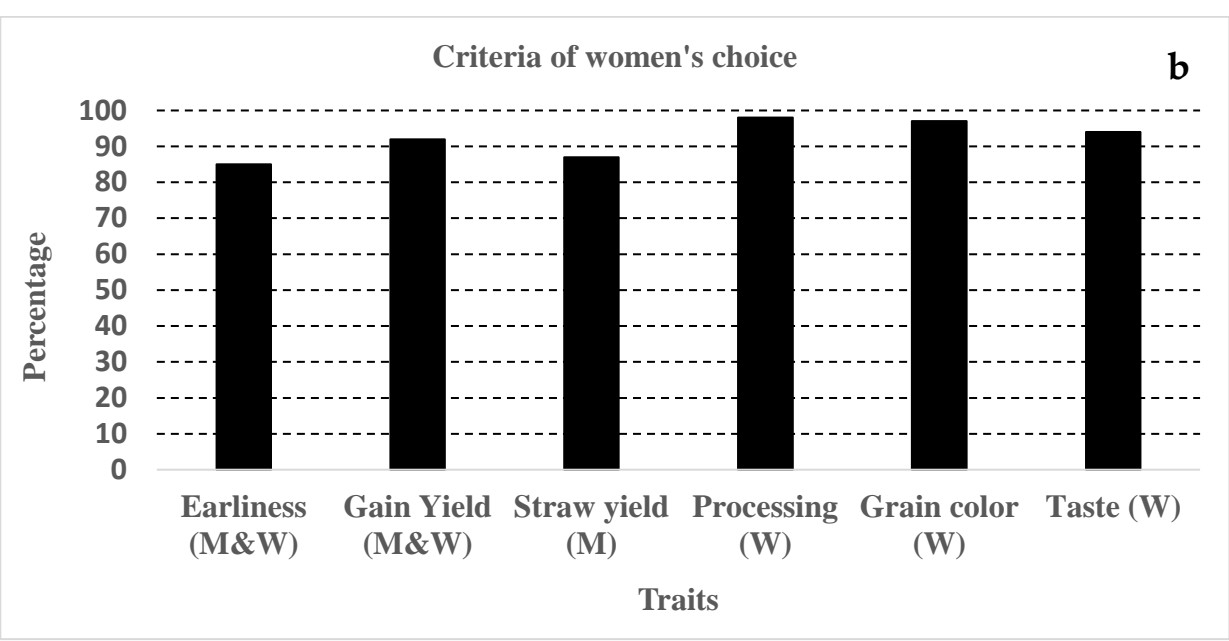

**Figure 5.** Criteria used by male (**a**) and female (**b**) farmers in Yaguine, Gory, and Foungou for selecting varieties, from the 2013/2014 to the 2015/2016 cropping seasons.

The PI gives an overall assessment of a variety. In the 2013/2014 season, men ranked the varieties from 10 to 82 (mean 44.6), whereas women gave a PI ranging from 17 to 78 (mean 49.3) (Table 9).

**Table 9.** Preference Index (%) of 54 sorghum varieties tested in Yaguiné, Gory, and Foungou in the 2013/2014 cropping season.

| Varieties | PI Men (%) | PI Women (%) | Mean PI (%) |
|---|---|---|---|
| Saba soto-25 | 74 | 62 | 68 |
| Saba soto | 31 | 26 | 29 |
| Saba soto-27 | 50 | 66 | 58 |
| 3009B | 16 | 35 | 25 |
| Magalèmè | 62 | 42 | 52 |
| Saba tienda-29 | 70 | 69 | 70 |
| CSM 63E | 10 | 27 | 19 |
| Saba tienda-31 | 48 | 73 | 61 |
| Modigiby | 64 | 44 | 54 |
| Saba tienda-32 | 37 | 42 | 40 |
| Saba soto-3 | 33 | 31 | 32 |
| Samé | 77 | 53 | 65 |
| Saba tienda-34 | 60 | 68 | 64 |
| Saba tienda-35 | 42 | 44 | 43 |
| Saba soto-12 | 16 | 33 | 25 |
| Lagahiré | 43 | 40 | 42 |
| Saba soto-13 | 18 | 42 | 30 |
| 12B | 10 | 17 | 13 |
| Boydibo | 61 | 51 | 56 |
| Saba soto-15 | 53 | 57 | 55 |
| Saba soto-16 | 28 | 47 | 37 |
| Saba soto-24 | 43 | 61 | 52 |
| Saba soto-32-1 | 75 | 72 | 73 |
| Lagahiré doumbé | 23 | 26 | 25 |
| Saba tienda-33 | 50 | 62 | 56 |

**Table 9.** *Cont.*

| Varieties | PI Men (%) | PI Women (%) | Mean PI (%) |
|---|---|---|---|
| Saba soto-4 | 47 | 66 | 56 |
| Saba soto-5 | 52 | 67 | 59 |
| Samba-nouha | 73 | 56 | 65 |
| Saba soto-20 | 43 | 47 | 45 |
| Saba soto-32-2 | 26 | 33 | 30 |
| Saba soto-8 | 80 | 73 | 77 |
| Saba soto-9 | 18 | 30 | 24 |
| Saba soto-10 | 20 | 34 | 27 |
| Saba soto-11 | 27 | 42 | 35 |
| Saba tienda-36 | 32 | 55 | 43 |
| Saba tienda-37 | 34 | 54 | 44 |
| Saba soto-22 | 40 | 52 | 46 |
| Saba tienda-39 | 25 | 36 | 30 |
| Malisor 92-1 | 15 | 42 | 28 |
| Saba soto-17 | 42 | 34 | 38 |
| Saba soto-18 | 34 | 40 | 37 |
| Saba soto-19 | 55 | 44 | 50 |
| Saba soto-6 | 74 | 51 | 62 |
| Saba soto-21 | 55 | 48 | 52 |
| Saba tienda-38 | 23 | 31 | 27 |
| Saba soto-23 | 82 | 74 | 78 |
| Saba soto-1 | 50 | 78 | 64 |
| Saba tienda-30 | 45 | 63 | 54 |
| Saba tienda | 70 | 55 | 62 |
| Saba soto-26 | 44 | 59 | 52 |
| Motafara | 55 | 56 | 55 |
| Saba soto-14 | 53 | 44 | 48 |
| Saba soto-28 | 40 | 49 | 45 |
| 82B | 60 | 63 | 61 |

PI = Preference index.

In general, there was good agreement between the rankings both men and women awarded to the varieties ($R^2 = 0.51$) (Figure 6). The varieties that had the highest average PI score (men and women combined) were Saba soto-23, Saba soto-8, Saba soto-32-1, Saba tienda-29, and Saba soto-25.

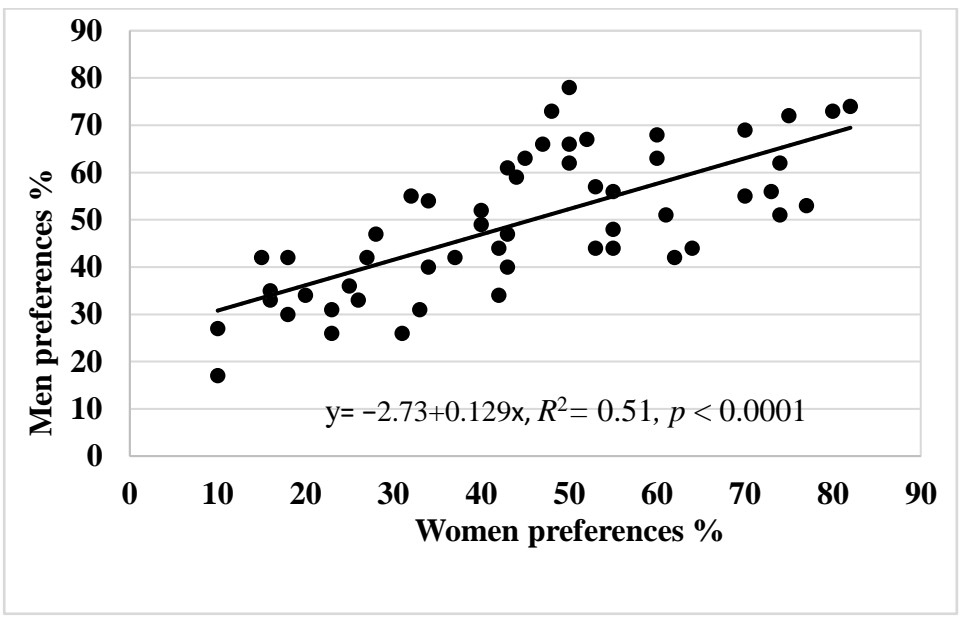

**Figure 6.** Women and men's ranking of varieties in the 2013/2014 season in Yélimané.



A ranking of the varieties was also undertaken among the 22 varieties that remained in the following year (2014/2015). The men gave these varieties PI scores ranging from 28 to 85, while the women assigned rankings from 30 to 81 (Table 10). The five top varieties ranked by men and women (combined) were Saba soto-23, Saba soto-8, Saba tienda-32-1, Saba tienda-31, and Saba tienda-29.

**Table 10.** Preference index of 22 sorghum varieties tested in Yaguiné, Gory, and Foungou in the 2014/2015 cropping season.

| Varieties | PI Men (%) | PI Women (%) | Mean PI (%) |
|---|---|---|---|
| 82B | 61 | 66 | 63 |
| Boidibo (local Yélimané) | 64 | 54 | 59 |
| Lagahiré (local Yélimané) | 46 | 43 | 45 |
| Magalèmè (local Yélimané) | 67 | 48 | 57 |
| Motafara (local Tombouctou) | 62 | 58 | 60 |
| Saba soto-10 | 35 | 43 | 39 |
| Saba soto-23 | 85 | 81 | 83 |
| Saba soto-25 | 71 | 61 | 66 |
| Malisor 92-1 | 56 | 49 | 52 |
| Saba soto-3 | 45 | 35 | 40 |
| Saba tienda-31 | 47 | 73 | 60 |
| Saba tienda-32-1 | 80 | 75 | 77 |
| Saba tienda-32-2 | 33 | 36 | 34 |
| Saba tienda-34 | 62 | 65 | 64 |
| Saba tienda-37 | 48 | 58 | 53 |
| Saba tienda-29 | 73 | 68 | 71 |
| CSM 63E | 28 | 30 | 29 |
| Saba soto-5 | 56 | 62 | 59 |
| Saba soto-8 | 83 | 79 | 81 |
| Saba soto-9 | 40 | 41 | 41 |
| Samba-Nouha (local Yélimané) | 75 | 59 | 67 |
| Samé (local Yélimané) | 74 | 56 | 65 |

PI = Preference index.

As in the previous season, there was a good agreement between men and women's rankings of the varieties ($R^2 = 0.61$) (Figure 7).

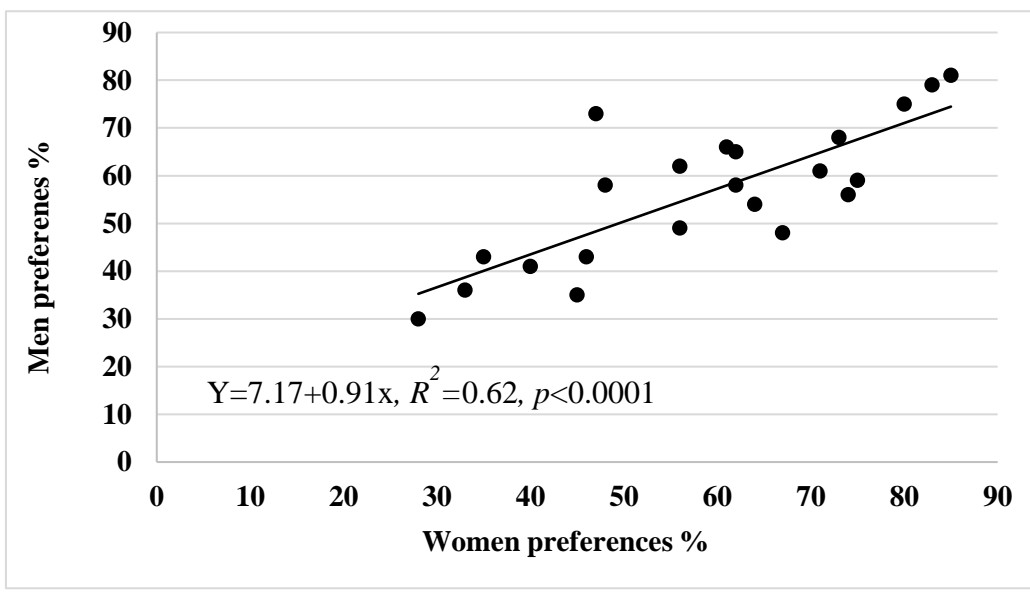

**Figure 7.** Women and men's rankings of varieties in the 2014/2015 season in Yélimané.

There was also a significant positive relationship between yield and the PI in the 2013/2014 season ($R^2 = 0.19$). The four varieties that had the highest yield were also those with the highest PI (Figure 8).

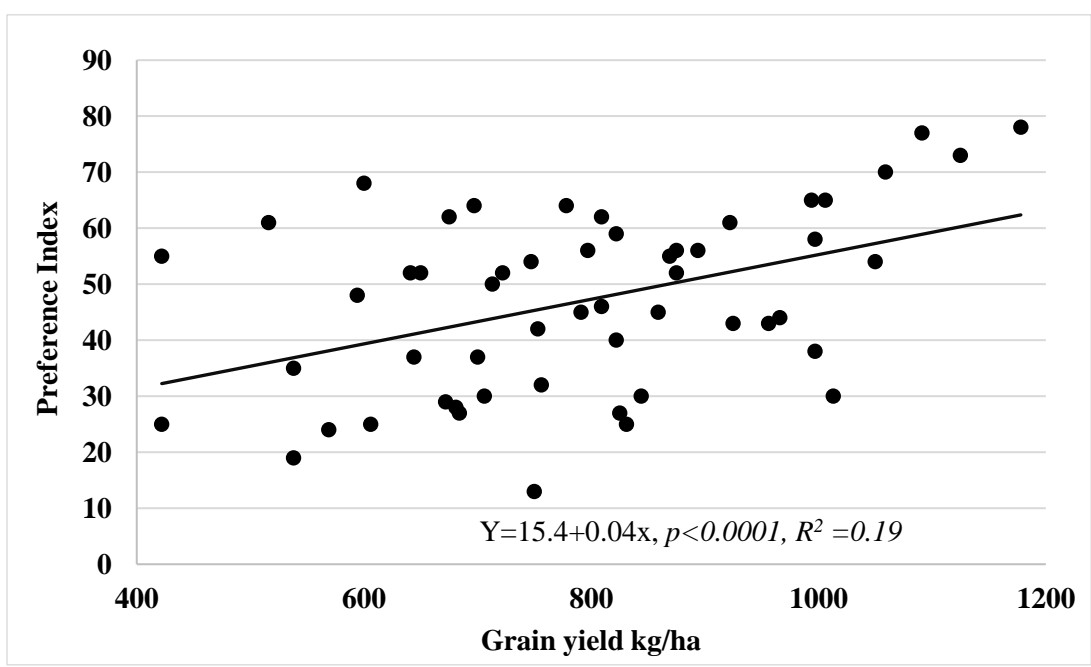

**Figure 8.** Relationship between preference index and yield in the 2013/2014 season in Yélimané.

As in 2013/2014, the varieties with the highest yield in 2014/2015 were among the preferred varieties according to the PI ($R^2 = 0.19$) (Figure 9).

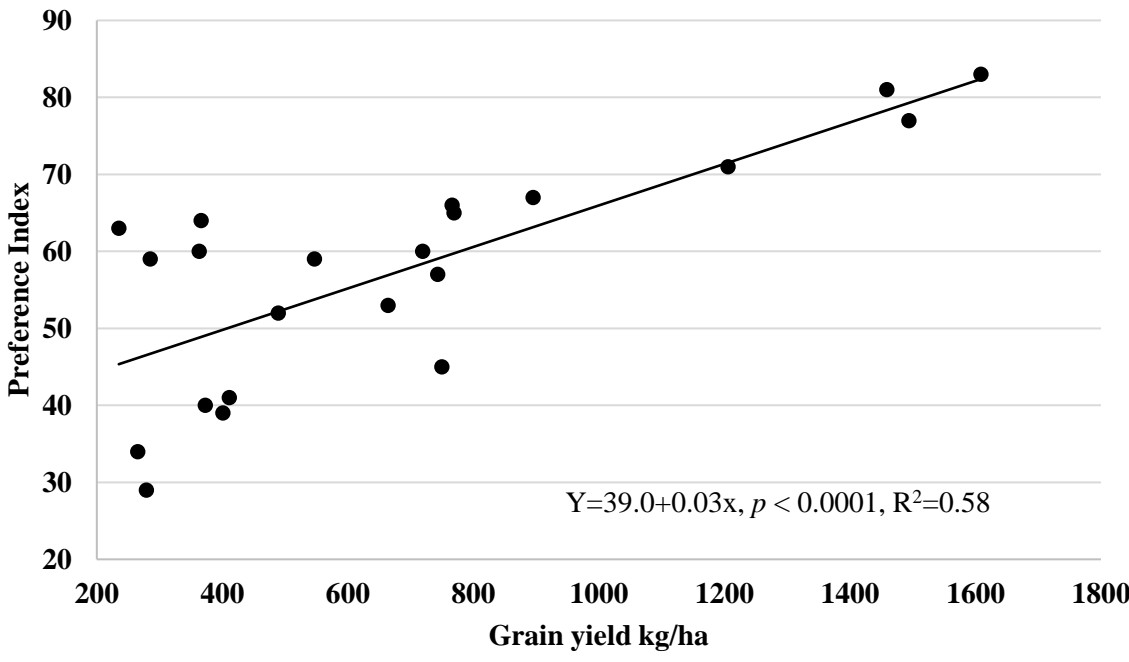

**Figure 9.** Relationship between grain yield and preference index in the 2014/2015 season.

### 3.5. Sorghum Seed System and Adoption Rate of Improved Varieties

A survey undertaken at the start of the project in 2013 showed that 70% of the farmers obtained their seeds from their own production. Seeds are procured through exchanges with other farmers in the same village, with farmers from different villages, and with

farmers during market days (Figure 10). The farmers maintain seeds of their preferred varieties from one cropping season to the another by storing them in a room (10%), on a string in a shelter (30%), or in a granary (60%).

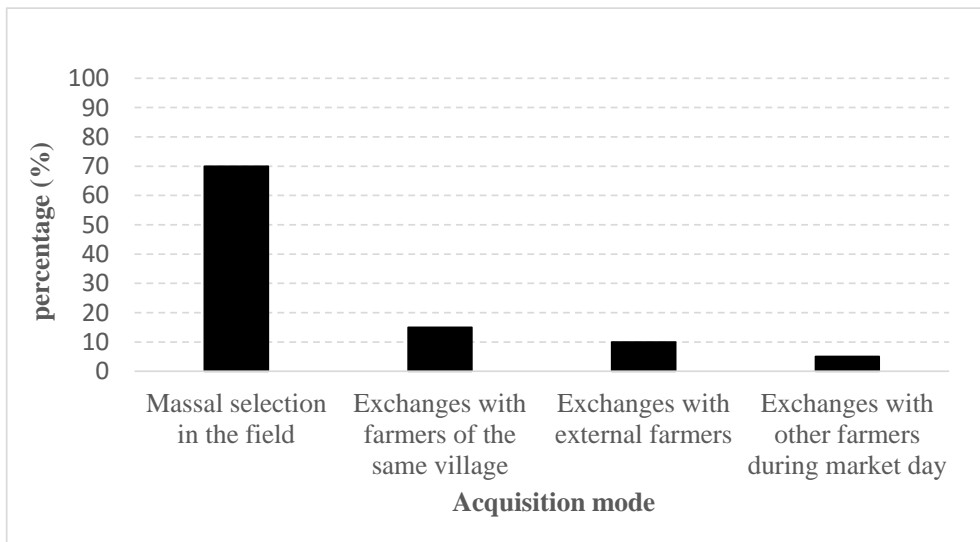

**Figure 10.** Mode of acquisition of sorghum seeds in flood recession farming in Yaguine, Gory, and Foungou from the 2013/2014 to the 2015/2016 cropping season.

After the four new sorghum varieties were developed and released in the area, all farmers, as they were doing before, continued to choose the best panicles for each variety separately and conserve seeds for the next cropping season. The varieties spread in the areas as farmers shared seeds with friends and family. The project team also provided seeds, but only for demonstration purposes in the farmers' field school. As of 2021, at least 50% of the farmers used two improved varieties or more, as shown in Figure 11. Only 3% of the respondents were not using improved varieties.

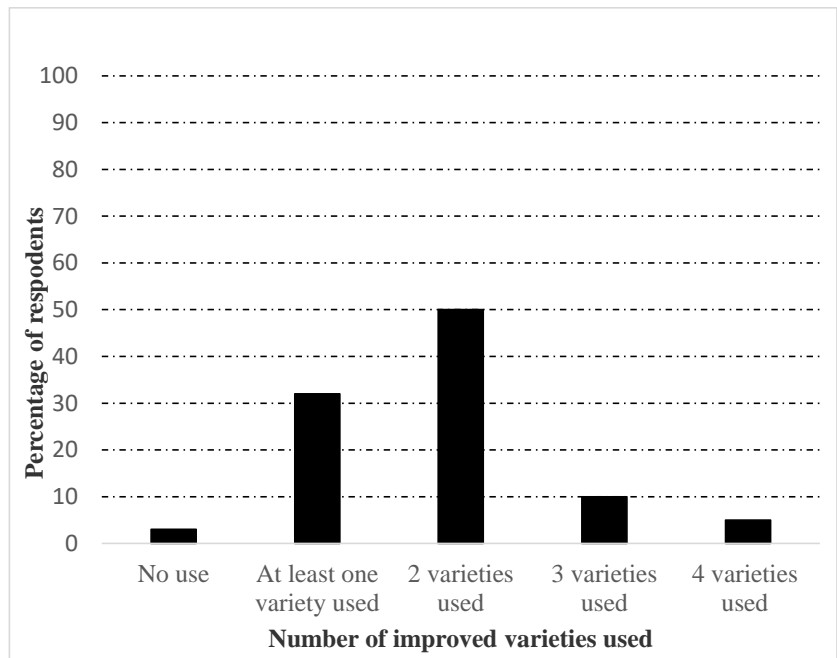

**Figure 11.** Use of improved sorghum varieties in Yaguine, Gory, and Foungou, from the 2013/2014 to the 2020/2021 cropping seasons.

The introduction of the new varieties in combination with higher crop density, seed treatment, and improved fertilization greatly improved food security among the 101 households included in the survey. Farmers who adopted the new production package used the technologies, on average, on 55% of the cultivated area. In comparing the number of food-insecure months for the three villages (101 households) before the implementation of the project (2013) and upon its completion (2021), we found an average reduction of 3.59 months (Table 11). The village with the highest adoption rate for the improved variety (Gory) also had the highest reduction in the number of food-insecure months.

**Table 11.** Impact of new flood recession sorghum varieties on household food security in Yaguine, Gory, and Foungou, from the 2013/2014 to the 2020/2021 cropping season.

| Sites | Food-Secure Months before Using Improved Sorghum Production Package | Food-Secure Months after Using Improved Sorghum Production Package | % of Improved Varieties Used |
|---|---|---|---|
| | Mean | Mean | Mean |
| Foungou($n$ = 23) | 6.33 | 9.44 | 33 c |
| Gory ($n$ = 38) | 5.67 | 10 | 50 a |
| Yaguiné ($n$ = 40) | 6.65 | 9.96 | 40 b |
| Mean | 6.22 b | 9.80 a | 41.00 |
| SD | 1.93 | 1.48 | 8.54 |
| CV (%) | 31.02 | 15.10 | 20.08 |
| Probability | 0.001 | | <0.001 |

Different letters below columns or after rows show statistically significant differences.

## 4. Discussion

Flood recession farming plays a central role in ensuring food security in the Yélimané area by supplementing rainfed food production. Previous studies have shown that yields can be doubled by introducing improved fertilization and crop-establishing methods like seed priming [1]. The current study shows that there is a great potential for increasing sorghum production by introducing improved varieties.

The traditional varieties differed greatly in time to maturity, plant height, seed color, and use in food preparation. Time from sowing to maturity varied from 75 to 180 days for the varieties studied. In the Yélimané area, the preferred local varieties took 90 days to reach 50% maturity, and they completed the growth cycle entirely on water stored in the soil profile. The two most commonly grown varieties are Samé and Magalèmè; these varieties are popular because of their palatability and versatility in food preparation.

The new varieties were tested against local varieties in three seasons. The varieties differed greatly in grain and stover yields in all three years. In 2013/2014, the yield varied from 422 to 1178 kg ha$^{-1}$, in 2014/2015 from 235 to 1609 ha$^{-1}$ (22 selected varieties from the previous year), and in 2015/2016, from 979 to 1608 kg ha$^{-1}$ (5 varieties). This shows that there is a lot to gain from using high-yield varieties.

The varieties with the highest yield were short maturing varieties that have a growth cycle from sowing to 50% maturity ranging from 55 to 65 days. The best-performing variety with regard to grain yield was Saba soto 23 (Yélimané 1); this variety provided 60.2% greater yield compared to the local variety (Samé) across three years. Furthermore, this variety also increased stover yield by 55.3% on average compared to Samé. Straw is of great value, as most households also keep livestock. Even in years when the grain harvest fails, the farmers can harvest some straw.

### 4.1. Relationship between Sorghum Traits

The relationships among eight traits of the tested varieties were assessed in a Pearson correlation analysis. The correlation coefficient was −0.84 between grain yield and time to maturity, while it was −0.30 for the relationship between stover yield and time to maturity. This shows that time to maturity is a stronger determining factor for grain yield than for

straw yield. The reason for this is that the amount of water stored in the soil profile is probably sufficient for the crop to complete vegetative growth (until flowering), but is insufficient to ensure grain filling. Late-maturing varieties will therefore be particularly penalized because there will not be enough water for them to complete the grain-filling period. Figure 4, on the moisture dynamics in the soil profile, also showed that there was very little water remaining in the soil profile beyond 80 days after sowing. This partly explains why the early maturing varieties (55–65-day cycle) performed better than the late varieties (90-day cycle). This also explains why the farmers rejected late-maturing varieties in Yélimané.

A positive correlation coefficient was found between grain yield and disease resistance. Such a relationship has been found previously in sorghum [13]. Furthermore, there was a positive correlation coefficient between grain yield and dehulling properties, grain pounding properties, and taste, showing that varieties with a high yield also have good post-harvesting properties. Late-maturing varieties may have less grain filling than early varieties, and this may negatively affect dehulling, pounding, and taste. Furthermore, there was a positive relationship among post-harvest properties. One reason for this is that a low degree of dehulling reduces the milling/pounding quality. Poor pounding properties can also increase mold infestation, which will again negatively affect taste. Furthermore, soft kernels are not easily dehulled, making milling difficult [10]. A low degree of dehulling may also affect the taste, as the bran (pericarp) tastes different than the rest of the grain (endocarp and embryo). Ease of dehulling is dependent on a high proportion of hard endosperm, thick pericarp, and no pigmented testa [10]. Milling quality depends on kernel size, shape, density, hardness, structure, the presence of a pigmented testa, pericarp thickness, and color. These factors may explain why there is a positive relationship among the post-harvesting properties.

A study in Mali on sorghum varieties in rainfed farming showed that farmers preferred early varieties, with open, droopy panicles, as these are less prone to diseases and bird attack, i.e., it is difficult for the birds to find a stable position in the panicles on these varieties [7]. Furthermore, farmers prefer varieties with open glumes at maturity, as they are more threshable. Varieties with hard grains are also preferred as they are easier to dehull and have better storage properties [7]. Hardness is also associated with a good taste. In addition, farmers prefer white grains over red and dark grains, as dark-colored grains contain more tannin [14]. Varieties with dark grains are also less attractive to birds. This shows that the post-harvest properties of the sorghum varieties in a flood recession setting are similar to those of the varieties used in rainfed cultivation.

### 4.2. Farmers' Ranking of Varieties

The PI for the varieties varied from 13 to 78 in the 2013/2014 season and from 34 to 83 in the 2014/2015 season. The variability in PI was lower in the 2013/2014 season compared to the 2014/2015 because only the best varieties were included in the latter season. In general, there was a good agreement between women and men's preferences regarding varieties, as the $R^2$ for 2013/2014 and 2014/2015 were 0.51 and 0.62 respectively. However, it was shown that women put more emphasis on post-harvesting properties than men. The positive relationship between grain yield and PI in the 2013/2014 and 2014/2015 seasons shows that the high-yielding varieties were also the ones preferred by the farmers. In both years, the four best-performing varieties with regard to grain yield were also those with the highest PI.

### 4.3. Use of the Improved Varieties

The four most preferred varieties were given the names Yélimané 1 (Saba Soto-23), Yélimané 2 (Saba soto-32), Yélimané 3 (Saba tiendra 8), and Yélimané 4 (Saba tiendra 29); these varieties were registered in 2016 in the regional Economic Community of West African States (ECOWAS) catalogue of sorghum varieties [15].

Farmers is Mali normally prefer photosensitive varieties in rainfed agriculture [16], but the four varieties selected in this study are only weakly photosensitive. The breeders minimized the photosensitivity of these sorghum varieties to develop earliness. Hence, photosensitivity is not a big issue with these varieties, since sowing is spread over time according to the recession of water. The local varieties used in flood recession farming are also weakly photosensitive.

The survey in the Yélimané area in 2021 showed that there was a good uptake of the improved varieties, and 50% of the farmers used at least two improved varieties (Figure 11). Farmers in Mali are used to keeping several sorghum varieties [17]. Those farmers that used improved varieties also used other improved technologies such as high plant density, seed priming, and microdosing. One important reason for the uptake of the improved varieties is that they reach maturity about one month earlier than traditional varieties. The earliness of the improved varieties has several advantages. For example, farmers can reap a harvest at a time of the year when food is in short supply. Furthermore, cultivating varieties with different times to maturity allows the farmers to spread the labor demand at harvest. In addition, using varieties with different times to maturity also reduces risk, as pests such as birds and locusts may attack at different times of the year. Farmers may also choose different varieties according to fodder requirements. Finally, farmers may prefer varieties based on processing properties and taste.

The formal sorghum seed sector is not functioning well. The sorghum network in Mali uses an approach characterized by the active participation of farmers/seed cooperatives in priority setting, selection of varieties, and the distribution/sale of varieties [17]. The development of sorghum for flood recession farming has used a similar participatory approach, but no seed co-operative or farmers' seed multiplication groups have been established in Yélimané. Farmers exchange seeds with their neighbors and within and between villages. There is no tradition in Mali for selling seeds to neighbors; the sharing of seeds is not considered a very efficient method to introduce new varieties in Mali [17]. The rapid spread of the new varieties can also be explained by the promotion of such varieties through local radio programs and by information exchange with other farmers through innovation platforms.

The varieties were tested at a moderate level of fertilization. To make good use of the varieties, they should be combined with a low-cost production approach consisting of a planting density of 66,000 pockets ha$^{-1}$, seed priming (8 h), seed treatment with a combined insecticide/fungicide, microdosing of NPK equivalent to 25 kg ha$^{-1}$, and microdosing of manure/compost equivalent to 500 kg ha$^{-1}$ [1]. This production technique has the potential to more than double yields compared to existing practices and increase yield stability due to improved crop establishment, reduced attacks by pests and diseases, and less exposure to end-of-season drought. We believe that such a low-cost production approach is not only of relevance to Mali, as flood recession farming is practiced in a band stretching from Senegal to Chad.

## 5. Conclusions

Through farmers' assessments and field experiments, the project was able to identify four improved sorghum varieties which are well-adapted to flood recession agriculture in the Yélimané district in Mali. These varieties were given the names Yélimané 1, Yélimané 2, Yélimané 3, and Yélimané 4, and were registered in 2016 in the ECOWAS catalogue of sorghum varieties. The improved varieties offer increased grain and stover yields, earlier maturity, and favorable post-harvesting properties. In the final year of the study, these varieties gave a grain yield advantage over the local variety by 64.2%, 50.7%, 49.0%, and 48.7%. The improved varieties were also the most preferred varieties according to assessments using the PI. They main difference of these varieties compared to the local varieties is that they can be harvested about one month earlier. Earliness is important in flood recession agriculture, as crop growth is entirely dependent on water stored in the soil profile. Furthermore, there was a positive correlation between high grain yield and

good post-harvesting processes and taste. Uptake of the varieties has been very good, and 5 years after the release of the varieties, more than 50% of the farmers continue to use at least two of them. Finally, households using the improved varieties in combination with improved production techniques have been able to reduce the number of food-insecure months by four.

**Author Contributions:** Conceptualization, K.T., G.S. and J.B.A.; methodology, K.T., B.T. and A.D.; validation, K.T.; formal analysis, K.T.; investigation, K.T., B.T. and A.D.; resources, J.B.A.; writing—original draft preparation, K.T.; writing—review and editing, G.S. and J.B.A.; supervision, K.T., G.S. and J.B.A.; project administration, K.T.; funding acquisition, J.B.A. All authors have read and agreed to the published version of the manuscript.

**Funding:** This research was funded by the Norwegian Ministry of Foreign Affairs.

**Acknowledgments:** We would like to thank the involved farmers in the Yélimané district for their participation. We greatly appreciate the cooperation of the field staff and collaborative technical services that participated in the project. We would also like to thank the Norwegian Ministry of Foreign Affairs for financing the Adapting Crop and Livestock Systems in Mali to Climate Change project.

**Conflicts of Interest:** The authors declare no conflict of interest.

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
