# Peer review of "Farmer Participatory Evaluation of Sorghum Varieties in Flood Recession Agriculture Systems in North-Western Mali"

_agronomy, doi:10.3390/agronomy12061379_

Round 1

Reviewer 1 Report

First of all I have to congratulate you on your work.

I would also like to make a few comments. 

1.Lines 143-145. Why do you choose to have just two rows? According to agricultural experimentation it is better to have more rows as to be able to collect your measures through the central rows in case to avoid any border effect.

2. It written that R2 below 90% results to is a good agreement. I am not so sure about that. I would prefer to change it if it is possible.

3. Finally, I think it would be nice if a general comparison could be made between the preferences of men and women for the whole (that is, adding everything and dividing by the whole). Stronger relationships are likely to emerge.

Author Response

Below you find our answer to reviewer1:

I would also like to make a few comments. 

1.Lines 143-145. Why do you choose to have just two rows? According to agricultural experimentation it is better to have more rows as to be able to collect your measures through the central rows in case to avoid any border effect.

Comment: There was limited space available for the experiment  since each experiment consisted of up to 54 varieties. This is the reason why  we had to restrict number of rows for each variety.

  1. It written that R2 below 90% results to is a good agreement. I am not so sure about that. I would prefer to change it if it is possible.

Comment: We are not sure what the reviewer mean here, but we consider a R2 above 90 as a very strong relationship.

  1. Finally, I think it would be nice if a general comparison could be made between the preferences of men and women for the whole (that is, adding everything and dividing by the whole). Stronger relationships are likely to emerge.

Comment: We have addressed this issue by better explaining the differences between men and women.

The revised text is follows:

Grain yield was almost unanimously the most important characteristic for both men and women , as 96% of them rated this property as important. The women tended to give more emphasis to processing properties, seed colour and taste than men. Men gave more emphasis to earliness than women.

Reviewer 2 Report

Recommendations for the authors

  • Line 215, Table 1: It seems that something is missing in third column, last row.
  • Line 226, 242, 268, 308, 316, 322: The letter p for the p-value should be italics (p = 0.05). Same applies for SD and CV.
  • Line 232, Figure: Please replace “R² = 0.6997” with “R² = 0.67”. Same applies to the rest of the figures.
  • Line 256, Figure 3: Please replace “p = < 0.0001” with “p < 0.0001”.
  • Line 324, Table 8: Please mark with “*” the statistically significant values (p<0.05)
  • Line 335-336, Figure 5: You may consider changing axis minor units to a greater value than zero. This way the differences with be stressed.
  • Line 348, Figure 6: Please replace “R2= 0.51, P<0.0001” with “R2= 0.51, p<0.0001”.
  • Line 359, Figure 7: Please replace “Y=7.17+0.91*X, R2=0.62, p<0.0001" with “y=7.17+0.91x, R2=0.62, p<0.0001”.
  • Line 360, Figure 7: I think a “%” is missing at x-axis? “Women preferences %”.
  • Line 369, Figure 8: Same as above. Figures 8, 9, and 10 are centered, while the rest figures have a right alignment.
  • Line 402, Table 11: Please check data for probability, something went wrong.
  • Line 403, Table 11: Please change the following to Italics: n, SD, CV.

Author Response

We have made the following changes based on comments from reviewers 2. The comments are marked in bold. 

Comment

  • Line 215, Table 1: It seems that something is missing in third column, last row.

Comment: We do not think anything is missing, but we have made some minor changes in the table by change some letter from capital to minor letters

  • Line 226, 242, 268, 308, 316, 322: The letter p for the p-value should be italics (p= 0.05). Same applies for SD and CV.

Comment: These changes have been made

  • Line 232, Figure: Please replace “R² = 0.6997” with “R² = 0.67”. Same applies to the rest of the figures.

Comment: These changes have been made.

  • Line 256, Figure 3: Please replace “p = < 0.0001” with “p < 0.0001”.

Comment: These changes have been made.

  • Line 324, Table 8: Please mark with “*” the statistically significant values (p<0.05)

Comment: We think this is sufficiently clear

  • Line 335-336, Figure 5: You may consider changing axis minor units to a greater value than zero. This way the differences with be stressed.

Comment: This change is not made because we think such a change may mislead many readers

  • Line 348, Figure 6: Please replace “R2= 0.51, P<0.0001” with “R2= 0.51, p<0.0001”.

Comment: This change is made

  • Line 359, Figure 7: Please replace “Y=7.17+0.91*X, R2=0.62, p<0.0001" with “y=7.17+0.91x, R2=0.62, p<0.0001”.

Comment: These changes are made.

  • Line 360, Figure 7: I think a “%” is missing at x-axis? “Women preferences %”.

Comment: This change is made.

  • Line 369, Figure 8: Same as above. Figures 8, 9, and 10 are centered, while the rest figures have a right alignment.

Comment: We think this can be fixed by the journal.

  • Line 402, Table 11: Please check data for probability, something went wrong.

Comment: The level of probability give (p<0.001) show the level of probability for a significant change in number of food insecure months before and after the introduction of the technology package. The probability given in the last column gives the probability for significant difference between the villages regarding to use of improved varieties

  • Line 403, Table 11: Please change the following to Italics: n, SD, CV.

Comment: This change is made.